# Reviving the sound of a 150-year-old insect: The bioacoustics of *Prophalangopsis obscura* (Ensifera: Hagloidea)

**Charlie Woodrow**[1], **Ed Baker**[2], **Thorin Jonsson**[3], **Fernando Montealegre-Z**[1]*

**1** School of Life Sciences, Joseph Banks Laboratories, University of Lincoln, Lincoln, United Kingdom,
**2** Natural History Museum, London, United Kingdom, **3** Institute of Biology, Karl-Franzens-University Graz, Graz, Austria

* fmontealegrez@lincoln.ac.uk

**Data Availability Statement:** All relevant data are within the paper and its Supporting Information files.

## Abstract

Determining the acoustic ecology of extinct or rare species is challenging due to the inability to record their acoustic signals or hearing thresholds. Katydids and their relatives (Orthoptera: Ensifera) offer a model for inferring acoustic ecology of extinct and rare species, due to allometric parameters of their sound production organs. Here, the bioacoustics of the orthopteran *Prophalangopsis obscura* are investigated. This species is one of only eight remaining members of an ancient family with over 90 extinct species that dominated the acoustic landscape of the Jurassic. The species is known from only a single confirmed specimen–the 150-year-old holotype material housed at the London Natural History Museum. Using Laser-Doppler Vibrometry, 3D surface scanning microscopy, and known scaling relationships, it is shown that *P. obscura* produces a pure-tone song at a frequency of ~4.7 kHz. This frequency range is distinct but comparable to the calls of Jurassic relatives, suggesting a limitation of early acoustic signals in insects to sonic frequencies (<20 kHz). The acoustic ecology and importance of this species in understanding ensiferan evolution, is discussed.

## Introduction

Acoustic communication systems have long been a popular model for understanding ecological and evolutionary relationships within and between species. In insects, acoustic systems for both signal generation and recognition have evolved a substantial variety of forms, to facilitate a range of communication functions [1, 2], offering many routes to studying the evolution of acoustic communication. However, for extinct and rare insect species, we are often limited in our abilities to infer details of specific communication systems, as we are unable to record the sounds such species generate or measure their hearing capabilities.

In katydids (or bush-crickets; Orthoptera: Ensifera) and their allies, pure-tone and broadband sound production has evolved as a key mechanism for mate attraction and conspecific recognition [3–6]. These sounds are produced by tegminal stridulation–the process of moving a hardened scraper on one forewing, against a row of teeth (the file) on the other, producing vibrations on the wing which are then amplified by specialized wing cells (namely the harp

**Funding:** This study was funded by a European Research Council grant no. ERCCoG-2017–773067 and an NSF - NERC grant no. NSF DEB-1937815 - NERC T014806/1 to FMZ.TJ is supported through the European Commission via a Horizon 2020 Marie Skłodowska-Curie fellowship (829208, InWingSpeak).

**Competing interests:** The authors have declared that no competing interests exist.

**Abbreviations:** $f_c$, carrier frequency; $f_o$, resonant frequency; LW, left wing; RW, right wing.

and mirror) [4, 5] to radiate sound. This mechanism of sound production is evolutionarily conserved across a majority of the Ensifera, and its characteristics have been understood since the early 1900s [7, 8]. The retention of this mechanism across a diverse range of taxa, and the increasing ability of state-of-the-art imaging and acoustic technologies, is rapidly allowing researchers to re-visit once inaccessible specimens with novel methodologies to advance our understanding of Ensiferan acoustic communication.

Here we investigate the bioacoustics of *Prophalangopsis obscura* (Walker, 1869) (Ensifera: Prophalangopsidae), an insect belonging to an ancient katydid family of over 90 known species dominant during the Jurassic, with only eight extant members [9]. The genus *Prophalangopsis*, formerly *Tarraga*, has remained monotypic ever since the discovery of *P. obscura*, and thus received considerable interest in relation to the evolutionary history of the Ensifera [10–12]. The enigmatic nature of the type specimen has been compounded by no further male specimens being discovered in over 150 years, and only 2 potential female specimens ever found [13]. In addition, no works have explored the ecology of this species due to their uncertain geographic distribution [13, 14]. Therefore, a thorough study of their acoustic capabilities could improve our understanding of the communication systems and acoustic ecology of *P. obscura* and its long extinct relatives [15–17], and aid in future rediscovery of the species.

Using micro-scanning Laser-Doppler Vibrometry (LDV), we reconstruct the vibration patterns and resonances of the sound production organs (forewings) of the *P. obscura* type specimen. Furthermore, we investigate the morphology of the stridulatory apparatus and tegmina in detail to compliment LDV experiments and infer the likely carrier frequency ($f_c$) of this species' song over 150 years after specimen preservation. Employing existing validated models, and novel measurements from LDV, we obtain $f_c$ for the acoustic signal of *P. obscura* and use morphological data to calculate acoustic signal structure. Using knowledge of the wing biomechanics of other extant members of this ancient family, we reconstruct the calling song of *P. obscura*, and discuss the importance of this species in understanding the evolution of ensiferan acoustic communication.

## Materials and methods

### The holotype material

*Prophalangopsis obscura* (Walker 1869) is a large orthopteran (~10 cm; tegmina wingspan) represented by a single specimen housed at the London Natural History Museum, South Kensington, UK (specimen NHMUK 013806185). Collection details are scarce, with the location information listed only as 'India'. The specimen was originally set in a resting position following collection, but sometime between 1898 and 1939, the specimen was re-mounted with both wings spread [10], a position which remains to this day (Fig 1). At an unknown time after 1939, the left foretibia was lost. The right foretibia, which contains the tympanic ear, remains intact (Fig 1C). Both forewings are present, with the stridulatory (sound producing) organs intact, however the left wing is torn along the apical axis (Fig 1A). In 2005, two female specimens identified as *P. obscura* were located in China, later published by Liu et al. [13]. While male specimens were not identified to confirm the identity of these specimens, they minimally belong to a close relative of *P. obscura*. No permits were required for the described study, which complied with all relevant regulations.

### Tegmina and stridulatory file anatomy

*P. obscura* possesses a stridulatory file on each forewing. The morphology of each file was imaged using an Alicona InfiniteFocus microscope (Bruker Alicona Imaging, Graz, Austria) at 20x objective magnification, resulting in one composite 3D-image of each file with a vertical

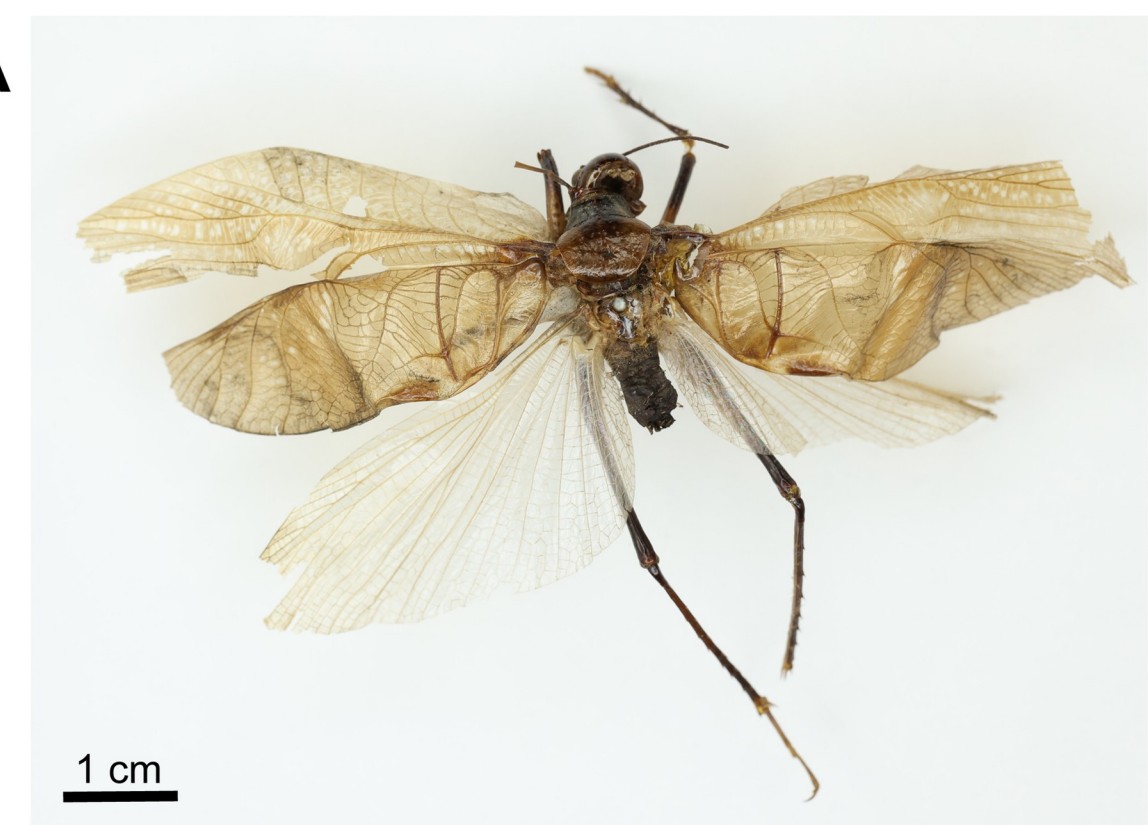

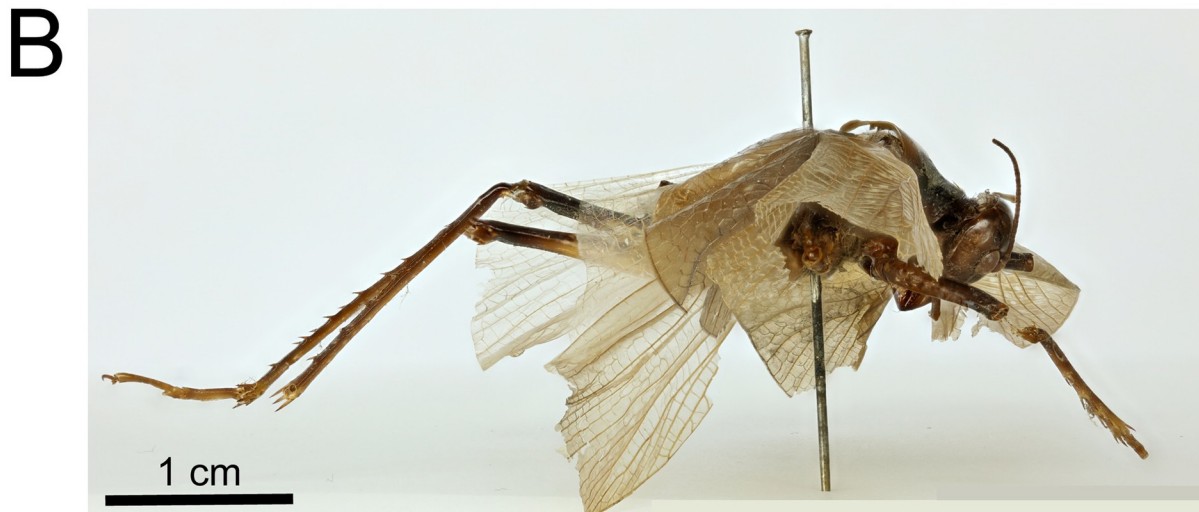

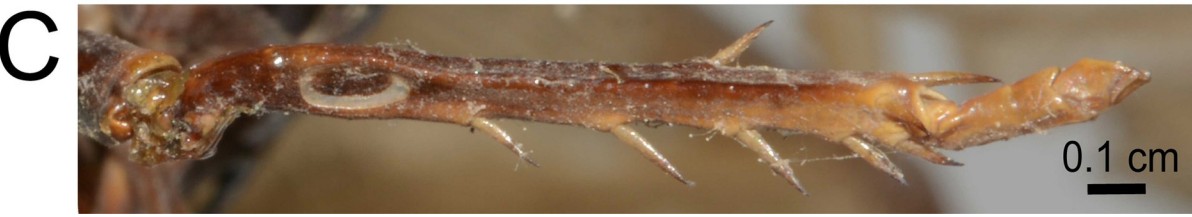

**Fig 1. The holotype of *Prophalangopsis obscura* (collected in India, Walker 1869).** A, dorsal habitus; B, lateral habitus; C, tympanal organ.

and horizontal resolution of 0.7 and 7.8 μm, respectively. Using the built-in Alicona software, the length of the stridulatory file was measured, as well as the spacing between stridulatory teeth (inter-tooth distances), and length of each tooth. The inter-tooth distance was measured as the distance between the central tip (cusp) of adjacent teeth.

### Forewing resonance and deflection pattern

The resonant frequency ($f_o$) and deflection patterns of the forewings was measured in the holotype of *P. obscura* using micro-scanning LDV (PSV-500, Polytec GmbH, Waldbronn, Germany), with approximately 1000 measuring points at a sampling frequency of 256 kHz. Acoustic signals for wing excitation were generated by the LDV internal data acquisition board (PCI-4451; National Instruments, Austin, TX, USA), and consisted of broadband periodic chirps ranging from 1 to 60 kHz at 60 dB SPL (re 20 μPa). The signal was amplified by a Pioneer A-400 amplifier (Pioneer, Kawasaki, Japan) and transmitted to a loudspeaker (Vifa, Avisoft Bioacoustics, Glienicke, Germany; flattened frequency response across the whole range) positioned 20 cm in front of the specimen. A reference signal to calculate the transfer function between the wing vibration and the stimulus was recorded using a 1/8" condenser microphone positioned horizontally at the wing plane between the wings (Model 4138-A015, Brüel & Kjaer, Nærum, Denmark). For further details of method, see [18].

### Reconstruction of the sound

To reconstruct the sound of *P. obscura*, several characteristics of the acoustic signal are required. These are (1) the song carrier frequency ($f_c$), (2) the decay of a single stridulatory tooth strike, (3) the number of oscillations produced during each stridulatory file strike (one full sound pulse), and (4) the timing between stridulatory file strikes. Previous investigations into the morphological parameters of katydid stridulatory apparatus have shown that the best predictors of the $f_c$ are the regions of mechanical displacement of the tegmina (the acoustically active wing cells), and the length of the stridulatory file [7]. In hagloids (Haglidae and Prophalangopsidae), which lack a specialised mirror area, it has been suggested that measurements of file length and LDV resonance, or the entire vibrating area, will be better predictors of $f_c$ [7, 8]. To predict $f_c$, we used existing models [7] to compare the frequency derived from the file length, right tegmen vibrational area, left tegmen vibrational area, and resonance from vibrometry.

Following calculation of the mean $f_c$ predicted by these four techniques, an artificial impulse of a single tooth strike of *P. obscura* was produced at this frequency, including a decay caused by damping. Oscillations of the tegmina mirror cells usually exhibit a free decay of 3–4 ms in species communicating at the determined $f_c$ [3, 11, 19], thus a 4 ms exponential decay was used.

Members of the Prophalangopsidae have a high stridulatory tooth density and short functional file length, which permits the generation of uniquely pure-tone calls [15, 20, 21], and as pure-tone singing katydids display a 1: 1 relation between tooth strikes and the number of oscillations in the song pulse [7], we used $f_c$, the number of functional stridulatory teeth, and the spacing of the teeth, to infer the pulse structure of the song of *P. obscura*. This was performed using a custom written Matlab code [15] which calculates the instantaneous period for each tooth impact based on the inter-tooth distance measurements. The resulting representative waveform of the acoustic signal of *P. obscura* was further analysed using the Signal Processing Toolbox in Matlab (R2021a, The MathWorks Inc., Natick, USA) with the following spectrogram parameters: FFT size 512, Hamming window, 50% overlap; frequency resolution: 512 Hz, temporal resolution: 0.15 ms.

## Results

### Tegmina and stridulatory file anatomy

The anatomy of the tegmina (forewings) of *P. obscura* is similar to those observed in both extant and extinct relatives of the Prophalangopsidae. The left wing (LW) and right wing (RW) display stridulatory files that are similar enough to be considered functionally symmetrical. The pattern of tooth distribution is slightly gaussian (Fig 2), suggesting the file could be adapted for sound production during the opening or closing phase of the wings. File length and number of teeth of LW and RW file were 9.60 and 9.99 mm and 134 and 137 teeth, respectively. The inter-tooth distances, tooth lengths, and plectra are symmetrical, suggesting both might have been capable of producing sound pulses (S1 Table).

### Forewing resonance and deflection pattern

Despite over 150 years of preservation, it was possible to obtain the deflection (vibratory) pattern of the forewings and $f_o$ in *P. obscura*. An assessment of the regions of the wings theoretically involved in resonant sound production and the displacement of the wings in response to an acoustic stimulus (Fig 3A and 3D) confirmed that the mirror and pre-mirror are the most likely regions for sound production in this species, as with all extant members of this family [21]. Displacement was highest within the mirror area for both the LW (Fig 3B) and RW (Fig 3E). The normalised displacement amplitudes of the mirror area of the LW displayed a peak frequency at 6.3 kHz (Fig 3C). However, despite morphological symmetry of the wings, the RW displayed a peak of 4.8 kHz (Fig 3F).

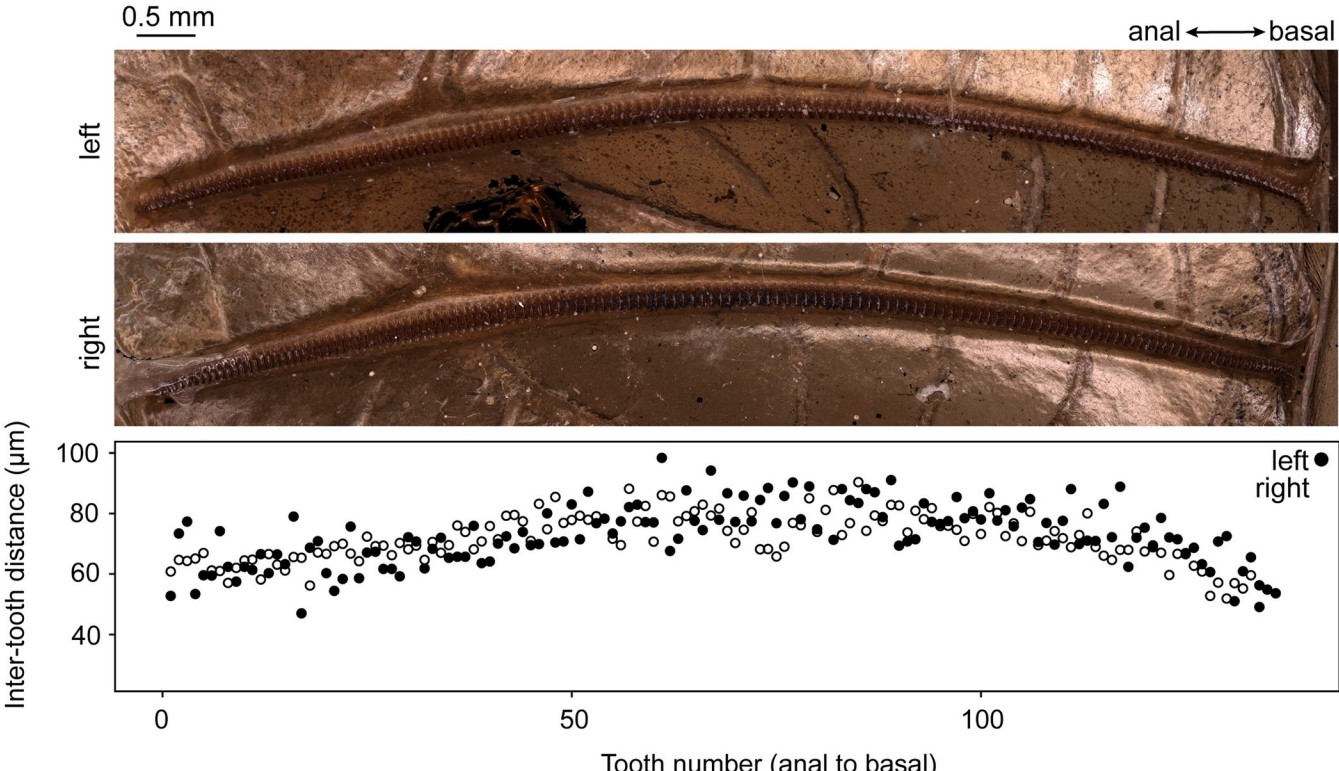

**Fig 2. Stridulatory file anatomy and inter-tooth distances in *Prophalangopsis obscura*.** Orientation of both files is along the anal (left) to basal (right) axis.

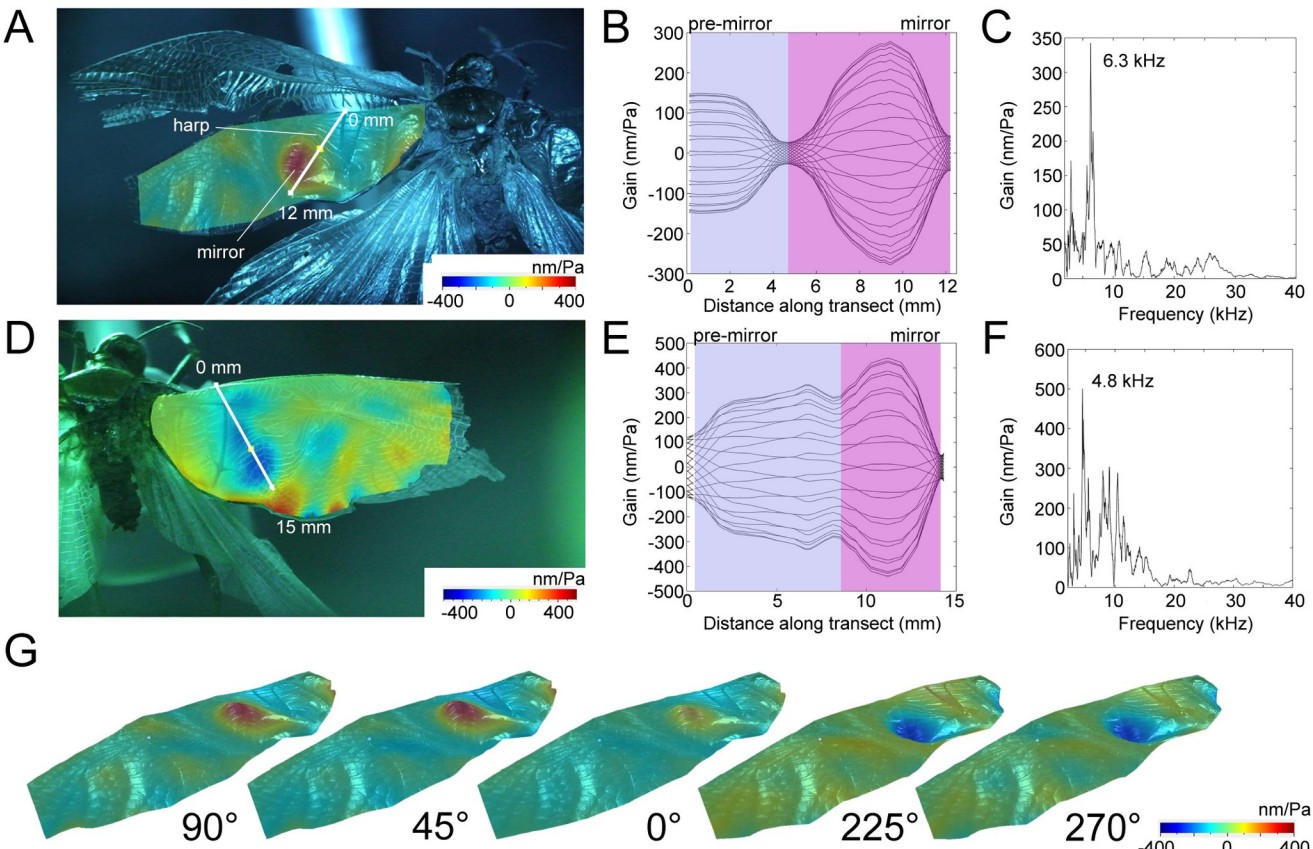

**Fig 3. Forewing resonance in *Prophalangopsis obscura*.** (A) Displacement map of the LW; (B) Deflection pattern of the white profile line in A; (C) Frequency spectrum of the left mirror; (D) Displacement map of the RW; (E) Deflection pattern of the profile line in D; (F) Frequency spectrum of the right mirror; (G) Angled view of the left forewing displacement pattern at 4.8 kHz.

### Reconstruction of the sound

Using $f_o$, stridulatory file length, and vibrational areas of the tegmina resulting from LDV deflection measurements, $f_c$ was calculated (Table 1). Based on phylogenetically controlled linear models of several measurement parameters [7], we believe the $f_c$ to be ~ 4.7 ± 0.05 kHz (Table 1). The measurements of inter-tooth distances and $f_c$ allowed the calculation of a time vector of a single sound pulse of the species' acoustic signal (For more details of the song reconstruction method, see [15]).

The Matlab script for sound pulse reconstruction [15] revealed that the structure of a single call pulse (Fig 4) is similar to that of fossil relatives of the same family [15], but differs in

**Table 1. Model measurement parameters for calculation of the likely carrier frequency ($f_c$) of *Prophalangopsis obscura*.**

| Measurement parameter | Measurement (x) | Slope (m) | Intercept (c) | $f_c$ (kHz) |
|---|---|---|---|---|
| File length (mm) | 9.6 | -0.97 | 3.74 | 4.693 |
| RW vibrating area (mm²) | 45.31 | -0.62 | 3.91 | 4.691 |
| LW vibrating area (mm²) | 39.05 | -0.54 | 3.53 | 4.716 |
| LDV resonance (kHz) | | | | 4.800 |
| Average | | | | 4.725 |

For all estimates of $f_c$: $\ln(f_c) = m * \ln(x) + c$, where ln = natural logarithm.

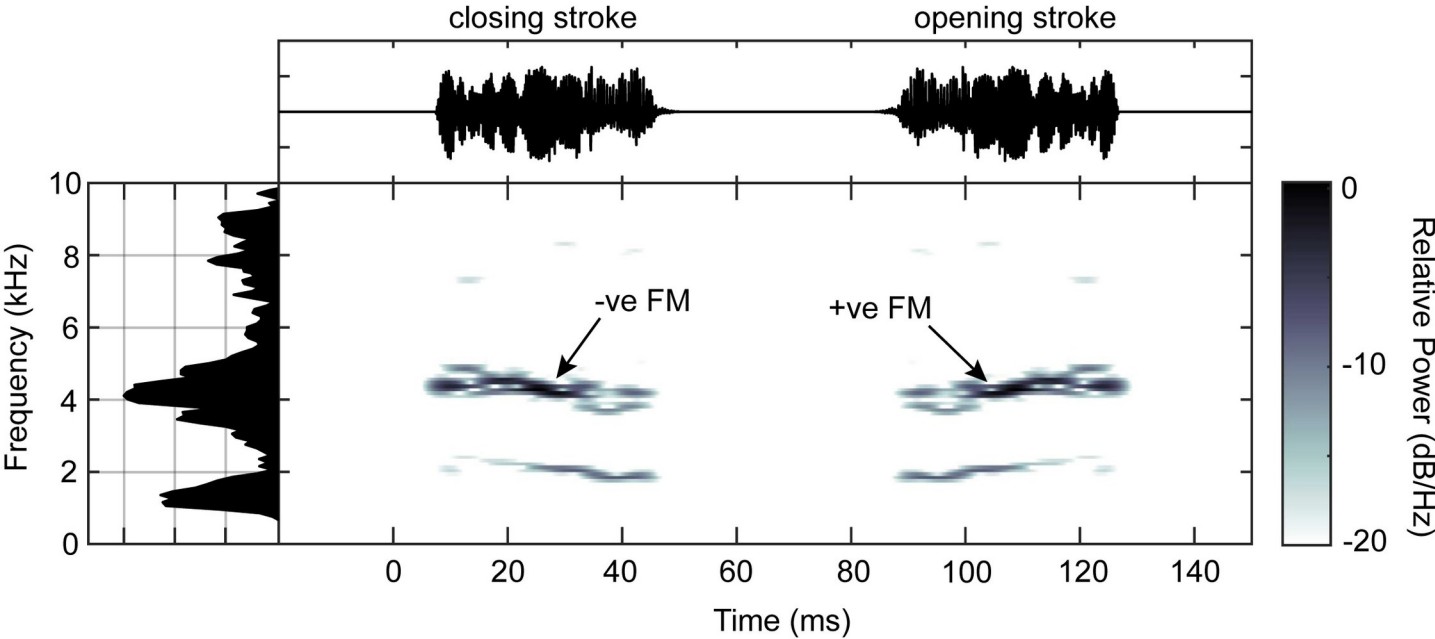

**Fig 4. Reconstruction and spectral analysis of a diplo-syllable containing two pulses of the sound of *Prophalangopsis obscura*.** Waveform of two chirps (top), with spectrogram below and frequency spectrum on the left marginal axis. The 2nd chirp is an artificial reversal of the 1st chirp, to demonstrate that frequency modulation (FM) will differ depending on whether sound is produced during the opening or closing wing stroke.

frequency and duration (Fig 4). The duration of a single pulse was found to be 42 ms (Fig 4), which is very close to the predicted pulse duration from functional file length using an existing model (40.78 ms; Montealegre-Z et al. 2017). Surprisingly, a slight frequency modulation in each chirp of the call was observed (Fig 4). Extant species display similar modulations as a result of changing velocity over the course of each wing stroke. The first predictions of how crickets produced their sounds looked at tooth distribution to infer whether sounds are produced during opening or closing of the wings [22]. As the frequency of the sound is a function of tooth strikes per time period, any changes to wing velocity over the course of one wing stroke will cause frequency modulation in the sound. Looking at the almost gaussian distribution of seemingly functional teeth in *P. obscura*, we cannot confirm whether this species is able to stridulate during the opening or closing of the wings, or both. The final reconstruction of the sound (Fig 4; S1 Audio) therefore consists of a putative diplo-syllable containing two pulses with every other chirp artificially reversed, to leave this element of the reconstruction open for future interpretations.

## Discussion

Using LDV techniques, we were able to obtain the deflection pattern of the tegmina of *P. obscura* and use information on tegmina and stridulatory file anatomy to reconstruct the song of the 150-year-old preserved museum specimen. The anatomy of the tegmina and stridulatory file display similarities to both fossil and extant prophalangopsids [15, 16, 21], and this similarity is also represented in the frequency and structure of the song (Fig 4), although $f_c$ here is lower than that of related fossil species [15]. The resonant frequency ($f_o$) of the tegmina provided by the LDV recordings matched the expected frequency from the models, and we were able to obtain the area of deflection, which was also used to calculate the potential calling song frequency (Table 1). Despite the matching frequency information provided by the right

tegmen, the left tegmen did not predict a similar $f_c$. We believe this discrepancy is due to a tear down the apical axis of the LW, given the similarities of the frequency predicted by the models to the resonance of the right tegmen (Table 1). In many singing ensiferans, the LW is found to be a better predictor of $f_c$ [7], however in the prophalangopsidae, it is known that the wings are functionally symmetrical [7], so we can be confident that the RW resonance is representative of the LW resonance. The retention of resonance in the RW may seem surprising, as insect cuticles become stiffer as they desiccate over time [23], and thus we may expect such stiffening to result in changes to resonance. However, in this case, we believe that the topology of the wings plays more of an important role in $f_o$, and due to the size of the tegmina, the effect of drying is not so pronounced. The thickness and area of the tegmina dictate the resonant properties of the musical areas of the wing, and larger musical areas display less variation in frequency response with changes to thickness [24]. For example, in the gryllid *Tarbinskiellus portentosus* with a harp size of ~25 mm$^2$, tegmina thickness would need to decrease by more than 30% before thickness would begin to greatly modify resonance [24]. Thus, for a large species like *P. obscura* which has a harp size of ~50 mm$^2$, even a significant change in tegmina thickness from desiccation would be unlikely to result in large changes to resonance, explaining why the resonance is here maintained. Nonetheless, further studies into the effects of wing thickness and tissue desiccation on tegmina resonance across orthopterans would offer a rich dataset for future works to calculate taxon-specific frequency changes over time, increasing the information we can obtain from dry museum specimens.

Just like the other extant members of this family, and unlike modern katydids (Tettigoniidae), *P. obscura* is likely capable of using both wings for singing, with both tegmina possessing symmetrical stridulatory files, plectra, and acoustically functional areas [15, 16, 21]. The mirror region of the tegmina displayed the greatest deflection, and the pattern of deflection followed that of extant relatives [21]. However, as suggested by Zeuner [10], the tegmina are not as specialised for sound production as other closely related extant species such as the great Grig *Cyphoderris monstrosa*. The size and function of the wings is one of the key features of *P. obscura* that separates it from the other extant prophalangopsids and resembles the specimens of the fossil record [10, 15]. While all other extant prophalangopsids (e.g. *Cyphoderris* spp.) are flightless and use their wings exclusively for sound production and mate attraction/gifting [25], *P. obscura* has wings potentially large enough for short or sustained flight, resembling both the extinct prophalangopsids and many tettigoniids.

Reduced flight is a well-established evolutionary mechanism to reduce or avoid predation by aerial predators, and in particular, bats [26]. The other extant species in this family, all of whom have lost the ability to fly, exhibit novel anti-predator defences, namely ultrasonic sound production organs [27], which likely evolved to act as a deterrent to a new host of predators they now face after switching to a terrestrial lifestyle. Such anti-predator adaptations are not present in *P. obscura*, nor are any other morphological adaptations associated with predation by bats such as enlarged cuticular spines [26]. We may predict therefore that this species lives in a region with reduced selection pressure from ultrasonic aerial predators, allowing it to retain the Jurassic form even after the emergence of echolocating bats [28]. Similarly, low frequency calling songs such as that of *P. obscura* are indicative of reduced pressures from eavesdropping predators, as low frequency sounds travel larger distances and could give away the location of the signaller [29]. Tettigoniids regularly predated by bats benefit from the increased attenuation of ultrasonic conspecific signals by a reduced detection range by eavesdropping predators [26]. However, it should be noted that correlating call frequency to ecology in such a manner does not consider other factors which will be driving call frequency evolution [26, 29].

Unfortunately, further inferences on natural history remain challenging as the precise origins of the type specimen remain obscure. Previous literature on the specimen references a

wide geographic area broadly synonymous with the extent of the former British India at the time of collection (e.g. Hindustan, E. Indies). The combined historical evidence suggests that the specimen was collected in northern India, although it is at present not possible to give a more precise location. If the female specimen described in Liu et al. [13] are confirmed to be *P. obscura* and not a closely related species, then the known range may be extended from northern India to include Tibet, a region certainly too cold to support an abundance of echolocating bats. Further collections from this area to confirm the association between males and females, and to investigate the local composition of potential predators, would be very valuable.

Following this song reconstruction, it may be plausible to deploy autonomous recording units (ARUs) into potential field sites and use signal detection algorithms to aid in the rediscovery of this species [30, 31]. We hope that in time, further specimens of Prophalangopsis obscura are located, to record the true song of this elusive species, and to validate the accuracy of the predictions presented here.

## Supporting information

**S1 Table. Morphological characters of the tegmina stridulatory files of P. obscura.**
(DOCX)

**S1 Audio. The reconstructed calling song of Prophalangopsis obscura.**
(WAV)

**S1 Fig. Visual reconstruction of Prophalangopsis obscura on a tree branch in a temperate montane habitat.** Illustrated by CW.
(JPG)

## Acknowledgments

Thank you to Dr Ben Price and NHM London for the loan of the specimen used in this study. Thanks also to the University of Lincoln's School of Life Sciences for CW's PhD studentship.

## Author Contributions

**Conceptualization:** Thorin Jonsson, Fernando Montealegre-Z.

**Data curation:** Charlie Woodrow, Thorin Jonsson.

**Formal analysis:** Charlie Woodrow.

**Funding acquisition:** Fernando Montealegre-Z.

**Investigation:** Charlie Woodrow, Ed Baker, Thorin Jonsson, Fernando Montealegre-Z.

**Methodology:** Charlie Woodrow, Thorin Jonsson, Fernando Montealegre-Z.

**Resources:** Ed Baker, Thorin Jonsson.

**Supervision:** Fernando Montealegre-Z.

**Visualization:** Charlie Woodrow.

**Writing – original draft:** Charlie Woodrow.

**Writing – review & editing:** Ed Baker, Thorin Jonsson, Fernando Montealegre-Z.

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
