## [Decision Letter · Decision Letter 0]

24 Mar 2022

PONE-D-22-01070Reviving the sound of a 150-year-old insect: the bioacoustics of Prophalangopsis obscura (Ensifera: Hagloidea)PLOS ONE

Dear Dr. Montealegre-Z,

Thank you for submitting your manuscript to PLOS ONE. After careful consideration, we feel that it has merit but does not fully meet PLOS ONE’s publication criteria as it currently stands. Therefore, we invite you to submit a revised version of the manuscript that addresses the points raised during the review process.

 Both reviewers appreciate the application of the techniques to museum specimens but had some minor points they would like you to address. It isn't necessary to address the issue of novelty for this journal, but it would be good to use controls or a discussion to address the issue of changing wing properties due to desiccation.

We look forward to receiving your revised manuscript.

Kind regards,

Vivek Nityananda

Academic Editor

PLOS ONE

Journal Requirements:

2. In your manuscript, please provide additional information regarding the specimens used in your study. Ensure that you have reported specimen numbers and complete repository information, including museum name and geographic location. If permits were required, please ensure that you have provided details for all permits that were obtained, including the full name of the issuing authority, and add the following statement: 'All necessary permits were obtained for the described study, which complied with all relevant regulations.' If no permits were required, please include the following statement: 'No permits were required for the described study, which complied with all relevant regulations.' For more information on PLOS ONE's requirements for palaeontology and archaeology research, see https://journals.plos.org/plosone/s/submission-guidelines#loc-paleontology-and-archaeology-research.

4. Thank you for stating the following in the Funding Section of your manuscript: 

"This study was funded by European Research Council Grant ERCCoG-2017-773067 and NSF-NERC grant 1937815 to FMZ. TJ is supported through the European Commission via a Horizon 2020 Marie Skłodowska-Curie fellowship (829208, InWingSpeak)."

We note that you have provided funding information. However, funding information should not appear in the Funding section or other areas of your manuscript. We will only publish funding information present in the Funding Statement section of the online submission form. 

"This study was funded by European Research Council Grant ERCCoG-2017-773067 and NSF-NERC grant 1937815 to FMZ. TJ is supported through the European Commission via a Horizon 2020 Marie Skłodowska-Curie fellowship (829208, InWingSpeak)."

Reviewers' comments:

Reviewer's Responses to Questions

**Comments to the Author**

1. Is the manuscript technically sound, and do the data support the conclusions?

Reviewer #1: Yes

Reviewer #2: Yes

2. Has the statistical analysis been performed appropriately and rigorously? 

Reviewer #1: N/A

Reviewer #2: Yes

3. Have the authors made all data underlying the findings in their manuscript fully available?

Reviewer #1: Yes

Reviewer #2: Yes

4. Is the manuscript presented in an intelligible fashion and written in standard English?

Reviewer #1: Yes

Reviewer #2: Yes

5. Review Comments to the Author

Reviewer #1: General: The authors model singing properties of a relict ensiferan Prophalangopsis obscura based on the preserved holotype specimen. They describe the properties of the stridulatory file and, using LDV, the resonant properties of the forewing, then reconstruct the species’ calling song using these data. P. obscura is an interesting study object, as it is one of the few extant hagloid species and represents a presumably early form even within this taxon, thus providing an evolutionarily basal snapshot of Ensifera in general. Using relatively recent methods such as LDV on a specimen that has been conserved over a century could facilitate further investigation of museum specimens of rare or extinct species. However, any control for the effect of drying on wing resonant properties are missing and likely can easily be added – though not of Prophalangopsis obscura.

Yet, the results are not very surprising. Quite similar experiments with closely related species have been previously published, partly by some of the authors themselves, which they also refer to (e.g. [21] Chivers et al. 2017, compare fig. 4 with fig. 4 of the current study). Therefore, the main novelty of the current manuscript lies in being the first to use LDV in a single conserved specimen instead of a living/fresh one. For making the data more valuable one needs some kind of control for the effects of wing desiccation. Reconstructing the song from the morphological and model data is a nice add-on. It would gain much value if a live male specimen were found (it seems that there might be living individuals in China).

Line 29: “…the bioacoustics of the orthopteran Prophalangopsis obscura are investigated.” Acoustics is treated as singular only when referring to the field of research in general.

Lines 35-36: The frequency range of extinct species is only reconstructed using similar models. Therefore, any comparison is confounded by the method itself.

Lines 72-74: The authors should mention the relatively recent discovery of possibly new specimens – even if only females; the relevant study is cited in the next sentence, but for a different reason altogether ([14] Liu et al. 2009).

Line 82: Bethoux (2012; ref. 13) in his Pl. 1CD in the appendix shows the results of exactly the same analysis – and the figure is basically identical to Fig. 2 in this manuscript. Therefore, it is not clear why you made this same effort a second time.

Line 83: The part after the semicolon is not a full sentence.

Line 96: The description of two females from China (caught 2005) by Liu et al. (2009) as Prophalangopsis obscura should be mentioned here. Even though one may claim that without a described male species identity is not demonstrated, these females in any case would belong to a very close relative.

Lines 120-121: see comment to line 82: Bethoux did not just define conventions but shows an identical reconstruction.

Lines 158-159: One could shortly mention the relevant arguments for that for the reader’s ease.

Paragraph 188 – 198: Measuring resonances of a completely dry wing is not what usually is done. The potential effects are not really discussed. While it obviously is not possible to make controls with P. obscura, one could rather easily make a kind of control with a long-winged bush cricket like Mecopoda elongata, which is reared in many labs. Also this species has low frequency components in its song and seeing whether and how the wing changes its resonances after drying would support (or contradict) arguments that the effect is rather minor. If you prefer a species with pure tone, then use a cricket like Gryllus or Teleogryllus. See your own discussion in lines 267-270.

Lines 221-222: already detailed in Methods.

Lines 261-262: That the LW is damaged is quite unfortunate, since usually the LW gave a better prediction than the RW ([7] Montealegre-Z et al. 2017). One should at least mention this here, since peak frequencies of LW and RW differ clearly. That RW resonance fits better to the expectations alone is a risky argument.

Line 280: If your sentence is true, it means that you question whether the species described by Liu et al. (2009) is even a prophalangopsid. Is that what you want to say? What are your arguments?

Lines 291-294: Solely relying on the “objective similarities between the external foretibial ear” without data is too speculative to suggest any functional similarities. Especially as it is not even clear which similarities the authors are referring to. Many bush cricket ears with open tympana look similar. The whole paragraph from line 284 on is rather general and speculative and should be shortened and merged with the following paragraph which mentions what is most desirable to get a better picture: find males of the females described from China.

Fig. 1: Consider using higher resolution photos.

Fig. 2: see above; identical to Bethoux (2012) Pl. 1CD; therefore, delimiting mirror, harp (and pre-mirror, if wished) in the wing in Fig. 4D (which could be shown larger in the foto) would be sufficient. Fig. 2 could go to supplementary materials.

Reviewer #2: This interesting MS reconstruct the sound of a rare and possibly extinct ensiferan by anatomical and biophysical measurements and models of the sound producing structures, finding that the call frequency fundamental is around 4.5 kHz. I think this is a well-conducted experiment and only have a few minor suggestions.

I suggest that you state the size of the animal somewhere in the text, although the reader can measure if from figure 1.

l. 146-148: I was a bit confused by this description: are you stating that the species here does not have a clearly defined mirror (although it is shown as a fairly well-defined area on fig 2) - btw you have not really defined what the mirror is at this stage in the MS.

l. 296-297: I think this sentence is unclear, suggest to rephrase as: 'allowing it to retain the Jurassic form even after the emergence of bats'

l. 302-304: The call frequency seems to be similar to the frequency of crickets, and it is also roughly the same size - are there similarities in lifestyles?

6. PLOS authors have the option to publish the peer review history of their article (what does this mean?). If published, this will include your full peer review and any attached files.

Reviewer #1: No

Reviewer #2: No

---

## [Author Response · Author response to Decision Letter 0]

5 May 2022

Line 29: “…the bioacoustics of the orthopteran Prophalangopsis obscura are investigated.” Acoustics is treated as singular only when referring to the field of research in general.

Response: Modified as suggested.

Lines 35-36: The frequency range of extinct species is only reconstructed using similar models. Therefore, any comparison is confounded by the method itself.

Response: Here we state the frequency range is similar to Jurassic extinct relatives. This comparison is brought about by the fact that ‘modern’ ensiferans (i.e. Tettigoniidae) are often communicating at frequencies exceeding the ultrasonic range, whereas the Jurassic species and P. obscura are using lower frequency signals like their extant hagloid relatives. We appreciate the concern that the model may produce similarities in the signals it produces, however the model used is an improved version of the one with which we compare (Gu et al., 2012), following Montealegre et al., 2017, for which a reconstruction such as this has not yet been produced. For both models, the frequency produced is very sensitive to the geometry and anatomy of the tegmina, and in this instance, the forewings are just very similar in shape and size to known Jurassic fossils, and so the frequency produced by P. obscura (4.7 kHz) is close to, for example, the Jurassic species A. musicus (6.4 kHz; Gu et al., 2012). The anatomy of the stridulatory file is similar in length to the fossil example, which is the most accurate predictor of calling song frequency (Montealegre-Z et al., 2017), and explains why the frequencies are similar. However these two frequencies are still very distinct, and if the model were to be repeated on an ultrasonic katydid for example, it would produce a very different signal. Future works in development by the senior author confirm that the model can be used to obtain a wide variety of calling signals from fossil and extant data, which will help to reassure readers that the model does not produce intrinsic similarities which render the discussions inappropriate. We have re-worded the comparisons to existing species to highlight more generally that this species is limited to the sonic range, and added ideas in the discussion r.e. the discovery of further fossil species to validate the sensitivity and potential problems of the model.

Lines 72-74: The authors should mention the relatively recent discovery of possibly new specimens – even if only females; the relevant study is cited in the next sentence, but for a different reason altogether ([14] Liu et al. 2009).

Response: Clarified here that there have been no further male specimens found, and that there are 2 female specimens that have been identified as P. obscura in the mentioned study. Also expanded as suggested in the discussion.

Line 82: Bethoux (2012; ref. 13) in his Pl. 1CD in the appendix shows the results of exactly the same analysis – and the figure is basically identical to Fig. 2 in this manuscript. Therefore, it is not clear why you made this same effort a second time.

Response: We decided to assess the wing topology based on the advancements of the field since the study of Bethoux in 2012, and to permit student training/understanding of the anatomy. We agree that the anatomy does not differ from that of Bethoux 2012, and thus we have also moved the wing description figure (Figure 2) into the supplemental materials.

Line 83: The part after the semicolon is not a full sentence.

Response: Semicolon removed.

Line 96: The description of two females from China (caught 2005) by Liu et al. (2009) as Prophalangopsis obscura should be mentioned here. Even though one may claim that without a described male species identity is not demonstrated, these females in any case would belong to a very close relative.

Response: We have included in this section the suggested information. We agree the specimens identified by Liu et al, if not P. obscura, are certainly a close relative.

Lines 120-121: see comment to line 82: Bethoux did not just define conventions but shows an identical reconstruction.

Response: We have moved the reconstruction to the supplemental material as it does not add any novel information on tegminal anatomy. This sentence has been removed from the methodology.

Lines 158-159: One could shortly mention the relevant arguments for that for the reader’s ease.

Response: Relevant arguments briefly added here as suggested to provide context for the reader.

Paragraph 188 – 198: Measuring resonances of a completely dry wing is not what usually is done. The potential effects are not really discussed. While it obviously is not possible to make controls with P. obscura, one could rather easily make a kind of control with a long-winged bush cricket like Mecopoda elongata, which is reared in many labs. Also this species has low frequency components in its song and seeing whether and how the wing changes its resonances after drying would support (or contradict) arguments that the effect is rather minor. If you prefer a species with pure tone, then use a cricket like Gryllus or Teleogryllus. See your own discussion in lines 267-270.

Response: We agree that a model to account for the changes in resonances of fresh vs. dry wings would expand this study into further assessments of museum specimens, and this is something that we are experimentally investigating currently as part of a different study. However any changes to wing resonance as a result of drying will be very dependent on topology, thickness, and size of the wings (Godthi et al., 2022), and thus a species like Mecopoda would dry in a different way to P. obscura, as the mirror region in Mecopoda is much smaller, more specialised, and thinner compared to the rest of the wing, whereas P. obscura does not demonstrate the same level of specialisation of the forewings for sound production. The thickness and area of the tegmina dictate the resonant properties of the musical areas of the wing, and larger musical areas display less variation in frequency response with changes to thickness (Godthi et al., 2022). For example, in the gryllid Tarbinskiellus portentosus with a harp size of ~25 mm2, tegmina thickness would need to decrease by more than 30% before thickness would begin to greatly modify resonance (Godthi et al., 2022). Thus, for a large species like P. obscura which has a harp size of ~50 mm2, even a significant change in tegmina thickness from desiccation would be unlikely to result in large changes to resonance. At this scale, tegmina size and topology is the key driver of resonance.

We believe from the information above and the matching of the wing resonance with the frequency suggested by the other methods used in the paper that the resonance has been maintained. We have expanded upon this validation in the discussion, but highlighted that for application of this method into the diverse world of museum collections requires generation of thorough validated models such as those by Godthi et al (2022) to reconstruct resonances across species.

Lines 221-222: already detailed in Methods.

Response: Sentences removed so not to repeat methodology unnecessarily.

Lines 261-262: That the LW is damaged is quite unfortunate, since usually the LW gave a better prediction than the RW ([7] Montealegre-Z et al. 2017). One should at least mention this here, since peak frequencies of LW and RW differ clearly. That RW resonance fits better to the expectations alone is a risky argument.

Response: The LW is a better predictor of calling song frequency in many ensiferans, which is mostly a result of the specialisation of the wings, whereby the left wing becomes the sound radiator and the right wing the scraper for more efficient sound radiation, such as in the Tettigoniidae, and the wings are restricted to a ‘left wing over right wing’ arrangement. However in the Haglidae, the wings and their resonances are symmetrical, and these species can switch between singing with their tegmina left over right, or right over left. In other extant hagloids in the mentioned paper, it is shown that sound frequency is dependent on the size of the vibrating area, and file which is symmetrical across wings. Thus, if the LW of P. obscura was not damaged, it would very likely suggest the same frequency information, particularly given that the LW vibrating area also predicted a similar frequency (Table 1).

Line 280: If your sentence is true, it means that you question whether the species described by Liu et al. (2009) is even a prophalangopsid. Is that what you want to say? What are your arguments?

Response: This section has been reworded during addressing of the next comment. We clarify that the specimens of Liu et al are either P. obscura, or a close relative.

Lines 291-294: Solely relying on the “objective similarities between the external foretibial ear” without data is too speculative to suggest any functional similarities. Especially as it is not even clear which similarities the authors are referring to. Many bush cricket ears with open tympana look similar. The whole paragraph from line 284 on is rather general and speculative and should be shortened and merged with the following paragraph which mentions what is most desirable to get a better picture: find males of the females described from China.

Response: We have removed speculative comments of comparisons in foretibial ear anatomy and shortened the paragraph as suggested. We believe that our inferences of the acoustic ecology of P. obscura, in particular in relation to predation pressures by bats, are valid hypotheses to remain in the discussion.

Fig. 1: Consider using higher resolution photos.

Response: New higher resolution images used for this figure. Panels showing stridulatory file have been removed, as these are shown in figure 2.

Fig. 2: see above; identical to Bethoux (2012) Pl. 1CD; therefore, delimiting mirror, harp (and pre-mirror, if wished) in the wing in Fig. 4D (which could be shown larger in the foto) would be sufficient. 

Response: Figure removed from main text and moved to supplementary materials. Areas of the wing added to figure 4 as suggested.

Fig. 2 could go to supplementary materials.

Figure removed from main text and moved to supplementary materials.

Reviewer #2: This interesting MS reconstruct the sound of a rare and possibly extinct ensiferan by anatomical and biophysical measurements and models of the sound producing structures, finding that the call frequency fundamental is around 4.5 kHz. I think this is a well-conducted experiment and only have a few minor suggestions.

Response: I suggest that you state the size of the animal somewhere in the text, although the reader can measure if from figure 1.

Specimen size briefly mentioned in the holotype material section of methods. As the specimen appears to have been eviscerated, which makes inferences of natural head to abdomen length challenging, we have reported wingspan as an indicator of size.

l. 146-148: I was a bit confused by this description: are you stating that the species here does not have a clearly defined mirror (although it is shown as a fairly well-defined area on fig 2) - btw you have not really defined what the mirror is at this stage in the MS.

Response: To clarify, we mean that although the region classed as a mirror based on venation is easy to identify, it is not as specialised for sound production as in the case of other ensiferan orthopterans. Clarity added to this section. Figure 2 has been moved to supplementary materials upon recommendation of reviewer 1.

Terminology for parts of the wing including mirror have been added to the introduction so the reader will be familiar with the term before this section.

l. 296-297: I think this sentence is unclear, suggest rephrasing as: 'allowing it to retain the Jurassic form even after the emergence of bats'

Response: Modified as suggested.

l. 302-304: The call frequency seems to be similar to the frequency of crickets, and it is also roughly the same size - are there similarities in lifestyles?

Response: The similarities in calling song frequency here definitely offers insights into predation pressures and ecology across the Ensifera. For example, crickets, as low frequency ground dwelling orthopterans, are under different predation pressures to katydids, which are often in vegetation and utilize much higher sound frequencies for communication. High frequency hearing and singing in katydids likely co-evolved (Song et al., 2020), maybe in response to the emergence of echolocating bats and the need to hear ultrasonic signals (Pulver et al., in review). Crickets on the other hand, not facing the predation pressure of bats due to their terrestrial lifestyle, have not faced such pressures to communicate at very high frequencies (although the Lebinthine crickets complicate the story). We may argue that this provides further evidence that P. obscura is not predated by bats, because as a large orthopteran clearly capable of flight, likely with a semi-arboreal lifestyle, we do not see adaptations for ultrasonic communication. However this argument is limited by a lack of information on hearing ranges in this species, and the presence of large, low frequency singing modern katydids such as Mecopoda, which are predated by bats. For the question you ask on similarities in the calling songs of crickets and P. obscura however, I believe this does not suggest similar lifestyles, just based on observations of the disparities in the lifestyles of similarly sized katydids and crickets.

---

## [Decision Letter · Decision Letter 1]

13 Jun 2022

Reviving the sound of a 150-year-old insect: the bioacoustics of Prophalangopsis obscura (Ensifera: Hagloidea)

PONE-D-22-01070R1

Dear Dr. Montealegre-Z,

We’re pleased to inform you that your manuscript has been judged scientifically suitable for publication and will be formally accepted for publication once it meets all outstanding technical requirements.

Kind regards,

Vivek Nityananda

Academic Editor

PLOS ONE

Additional Editor Comments (optional):

One author also had a very minor edit which you might want to incorporate.

Reviewers' comments:

Reviewer's Responses to Questions

**Comments to the Author**

1. If the authors have adequately addressed your comments raised in a previous round of review and you feel that this manuscript is now acceptable for publication, you may indicate that here to bypass the “Comments to the Author” section, enter your conflict of interest statement in the “Confidential to Editor” section, and submit your "Accept" recommendation.

Reviewer #1: All comments have been addressed

Reviewer #2: (No Response)

2. Is the manuscript technically sound, and do the data support the conclusions?

Reviewer #1: Yes

Reviewer #2: Yes

3. Has the statistical analysis been performed appropriately and rigorously? 

Reviewer #1: N/A

Reviewer #2: Yes

4. Have the authors made all data underlying the findings in their manuscript fully available?

Reviewer #1: Yes

Reviewer #2: Yes

5. Is the manuscript presented in an intelligible fashion and written in standard English?

Reviewer #1: Yes

Reviewer #2: Yes

6. Review Comments to the Author

Reviewer #1: Thank you for addressing all our comments and concerns adequately and explaining all the subjects in detail!

Reviewer #2: I think the authors have responded adequately to the suggestions by me and the other referee.

Just one minor clarification:

Methods, l 103 states that: 'The right foretibia, which contains the tympanic ear, remains intact'

suggest to change to: 'The right foretibia, which contains the right tympanic ear, remains intact'

7. PLOS authors have the option to publish the peer review history of their article (what does this mean?). If published, this will include your full peer review and any attached files.

Reviewer #1: No

Reviewer #2: No

---

## [Editor Report · Acceptance letter]

17 Jun 2022

PONE-D-22-01070R1 

Reviving the sound of a 150-year-old insect: the bioacoustics of *Prophalangopsis obscura* (Ensifera: Hagloidea) 

Dear Dr. Montealegre-Z:

I'm pleased to inform you that your manuscript has been deemed suitable for publication in PLOS ONE. Congratulations! Your manuscript is now with our production department. 

Kind regards, 

on behalf of

Dr. Vivek Nityananda 

Academic Editor

PLOS ONE